# Challenges and Opportunities in High-dimensional Variational Inference

**Akash Kumar Dhaka***
Aalto University, Silo AI
akash.dhaka@aalto.fi

**Alejandro Catalina***
Aalto University
alejandro.catalina@aalto.fi

**Manushi Welandawe**
Boston University
manushi.welandawe@bu.edu

**Michael Riis Andersen**
Technical University of Denmark
miri@dtu.dk

**Jonathan H. Huggins**
Boston University
huggins@bu.edu

**Aki Vehtari**
Aalto University
aki.vehtari@aalto.fi

## Abstract

Current black-box variational inference (BBVI) methods require the user to make numerous design choices—such as the selection of variational objective and approximating family—yet there is little principled guidance on how to do so. We develop a conceptual framework and set of experimental tools to understand the effects of these choices, which we leverage to propose best practices for maximizing posterior approximation accuracy. Our approach is based on studying the pre-asymptotic tail behavior of the density ratios between the joint distribution and the variational approximation, then exploiting insights and tools from the importance sampling literature. Our framework and supporting experiments help to distinguish between the behavior of BBVI methods for approximating low-dimensional versus moderate-to-high-dimensional posteriors. In the latter case, we show that mass-covering variational objectives are difficult to optimize and do not improve accuracy, but flexible variational families can improve accuracy and the effectiveness of importance sampling—at the cost of additional optimization challenges. Therefore, for moderate-to-high-dimensional posteriors we recommend using the (mode-seeking) exclusive KL divergence since it is the easiest to optimize, and improving the variational family or using model parameter transformations to make the posterior and optimal variational approximation more similar. On the other hand, in low-dimensional settings, we show that heavy-tailed variational families and mass-covering divergences are effective and can increase the chances that the approximation can be improved by importance sampling.

## 1 Introduction

A great deal of progress has been made in black-box variational inference (BBVI) methods for Bayesian posterior approximation, but the interplay between the approximating family, divergence measure, gradient estimators and stochastic optimizer is non-trivial – and even more so for high-dimensional posteriors [1, 10, 29, 31]. While the main focus in the machine learning literature has

---

*Equal contribution.

35th Conference on Neural Information Processing Systems (NeurIPS 2021).

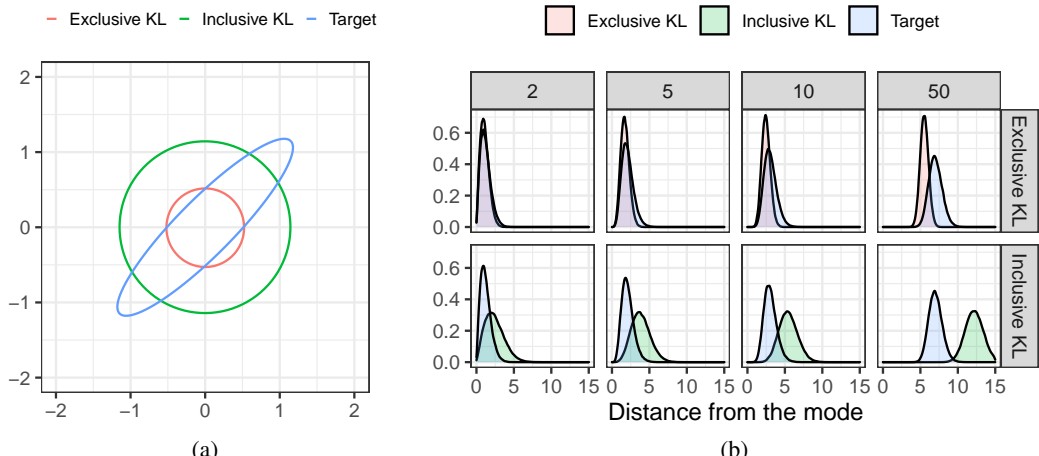

Figure 1: Illustration of a mean-field approximation with exclusive (mode-seeking) and inclusive (mass-covering) divergences. **(a)** The typical 2D illustration (correlation 0.9) gives the impression that the inclusive divergence would provide a better approximation. **(b)** For correlated Gaussian targets in dimensions $D = 2, 5, 10, 50$, the marginal distributions of the distance from the mode for samples drawn from the approximation (red) and the target (blue). The intuition from the low-dimensional examples does not carry over to higher dimensions: although the importance ratios are still bounded, even for a lower correlation level, the overlap in typical sets of the target and the approximations gets worse both for exclusive and inclusive divergences.

been on improving predictive accuracy, the choice of method components becomes even more critical when the goal is to obtain accurate summaries of the posterior itself.

In this paper, we show that, while the choice of approximating family and divergence is often motivated by low-dimensional illustrations, the intuition from these examples do not necessarily carry over to higher-dimensional settings. By drawing a connection between importance sampling and the estimation of common divergences used in BBVI, we are able to develop a comprehensive framework for understanding the reliability of BBVI in terms of the *pre-asymptotic* behavior of the density ratio between the target and the approximate distribution. When this density ratio is heavy-tailed, even unbiased estimators exhibit a large bias with high probability, in addition to high variance. Such heavy tails occur when there is a mismatch between the typical sets of the approximating and target distributions. In higher dimensions, even over-dispersed distributions miss the typical set of the target [18, 27]. Thus, as illustrated in Fig. 1, the benefits of heavy-tailed approximate families and divergences favoring mass-covering diminish as dimensionality of the target distribution increases. Building on these insights, we make the following main contributions:

1. We develop a conceptual and experimental framework for predicting and empirically evaluating the reliability of BBVI based on the choice of variational objective, approximating family, and target distribution. Our framework also incorporates the Pareto $k$ diagnostic [27] as a simple and practical approach for obtaining empirical and conceptual insights into the pre-asymptotic convergence rates of estimators of common divergences and their gradients.

2. We validate our framework through an extensive empirical study using simulated data and many commonly used real datasets with both Gaussian and non-Gaussian target distributions. We consider the exclusive and inclusive Kullback-Leibler (KL) divergences [4, 21, 24], tail-adaptive $f$-divergence [29], $\chi^2$ divergence [7], and $\alpha$-divergences [12], and the resulting variational approximation for isotropic Gaussian and Student-$t$ and normalising flow families.

3. Based on our framework and numerical results, we provide justified recommendations on design choices for different scenarios, including low- to moderate-dimensional and high-dimensional posteriors.

## 2 Preliminaries and Background

Let $p(\theta, Y)$ be a joint distribution of a probabilistic model, where $\theta \in \mathbb{R}^D$ is a vector of model parameters and $Y$ is the observed data. In Bayesian analysis, the posterior $p(\theta) := p(\theta \mid Y) = p(\theta, Y)/p(Y)$ (where $p(Y) := \int p(\theta, Y)d\theta$) is typically the object of interest, but most posterior summaries of interest are not accessible because the normalizing integral, in general, is intractable. Variational inference approximates the exact posterior $p(\theta \mid Y)$ using a distribution $q \in \mathcal{Q}$ from a family of tractable distributions $\mathcal{Q}$. The best approximation is determined by minimizing a divergence $D(p \parallel q)$, which measures the discrepancy between $p$ and $q$:

$$q_{\lambda^*} = \arg \min_{q_\lambda \in \mathcal{Q}} D(p \parallel q), \tag{1}$$

where $\lambda \in \mathbb{R}^K$ is a vector parameterizing the variational family $\mathcal{Q}$. Thus, the properties of the resulting approximation $q$ are determined by the choice of variational family $\mathcal{Q}$ as well as the choice of divergence $D$.

The family $\mathcal{Q}$ is often chosen such that quantities of interest (e.g., moments of $q$) can be computed efficiently. For example, $q$ can be used to compute Monte Carlo or importance sampling estimates of the quantities of interest. Let $w(\theta) := p(\theta, Y)/q(\theta)$ denote the density ratio between the joint and approximate distributions. For a function $\phi : \mathbb{R}^D \to \mathbb{R}$, the biased *self-normalized importance sampling estimator* for the posterior expectation $\mathbb{E}_{\theta \sim p}[\phi(\theta)]$ is given by

$$\hat{I}(\phi) := \sum_{s=1}^{S} \frac{w(\theta_s)}{\sum_{s'=1}^{S} w(\theta_{s'})} \phi(\theta_s),$$

where $\theta_1, \ldots, \theta_S \sim q$ are independent. Using importance sampling can allow for computation of more accurate posterior summaries and to go beyond the limitations of the variational family. For example, it is possible to estimate the posterior covariance even when using a mean-field variational family.

**Pareto Smoothed Importance Sampling.** Since importance sampling estimates can have very high variance, Pareto smoothed importance sampling (PSIS) can be used to substantially reduce the variance with small additional bias [27]. This procedure modifies and stabilises extreme importance ratios using a generalized Pareto distribution fit to the upper tail of the distribution of the ratios.

**Variational families.** Let $q_\lambda(\theta)$ be an approximating family parameterised by a $K$-dimensional vector $\lambda \in \mathbb{R}^K$ for $D$-dimensional inputs $\theta \in \mathbb{R}^D$. Typical choices of $q$ include mean-field Gaussian and Student's $t$ families [3, 14], full and low rank Gaussians [15, 22], mixtures of exponential families [17, 19], and normalising flows [25]. We focus on the most popular mean-field and normalizing flow families. Mean-field families assume independence across the $D$ dimensions: $q(\theta) = \prod_{i=1}^{D} q_i(\theta_i)$, where each $q_i$ typically belongs to some exponential family or other simple class of distributions. Normalising flows [1] provide more flexible families that can capture correlation and non-linear dependencies. A normalizing flow is defined via the transformation of a probability density through a sequence of invertible mappings. By composing several maps, a simple distribution such as a mean-field Gaussian can be transformed into a more complex distribution [25].

$f$**-divergences.** The most commonly used divergences are examples of $f$-divergences [28]. For a convex function $f$ satisfying $f(1) = 0$, the $f$-divergence is given by

$$D_f(p \parallel q) := \mathbb{E}_{\theta \sim q} \left[ f \left( \frac{p(\theta \mid Y)}{q(\theta)} \right) \right].$$

The exclusive Kullback-Leibler (KL) divergence corresponds to $f(w) = -\log(w)$, the inclusive KL divergence corresponds to $f(w) = w \log(w)$, the $\chi^2$ divergence corresponds to $f(w) = (w-1)^2$, and the general $\alpha$-divergences correspond to $(w^\alpha - w)/\{\alpha(\alpha - 1)\}$. We also consider the *adaptive $f$-divergence* proposed by Wang et al. [29].

**Loss estimation and stochastic optimization.** In all the cases we consider, minimizing the $f$-divergence is equivalent to minimizing the loss function $\mathcal{L}_f(p \parallel q) := \mathbb{E}_{\theta \sim q}[f(w(\theta))]$ (although, see Wan et al. [28] for a different approach). Let $L(\lambda) := \mathcal{L}_f(p \parallel q_\lambda)$ denote the loss as a function of the variational parameters $\lambda$. The loss and its gradient $G(\lambda) := \nabla_\lambda L(\lambda)$ can both be approximated using, respectively, the Monte Carlo estimates

$$\widehat{L}(\lambda) = \tfrac{1}{S} \sum_{s=1}^{S} f(w(\theta_s)) \quad \text{and} \quad \widehat{G}(\lambda) = \tfrac{1}{S} \sum_{s=1}^{S} g(\theta_s), \tag{2}$$

where $\theta_1, \ldots, \theta_S$ are independent draws from $q_\lambda$ and $g : \mathbb{R}^K \to \mathbb{R}^K$ is an appropriate gradient-like function that depends on $f$ and $w$. The two most popular gradient estimators in the literature are the score function and the reparameterization gradient estimator [20, 30]. The *score function gradient* corresponds to $g(\theta) = \{f(w(\theta)) - w(\theta)f'(w(\theta))\}\nabla_\lambda \log q_\lambda(\theta)$. It is a general-purpose estimator that applies to both discrete and continuous distributions $q$, but it is known to suffer from high variance. When this estimator is used for the *mass-covering* divergences such as the inclusive KL and general $\alpha$-divergences with $\alpha > 1$, the importance weights are usually replaced with self-normalized importance weights $w(\theta_s)/\sum_{i=1}^S w(\theta_i)$. The *reparameterization gradient* [20] requires expressing the distribution $q_\lambda$ as a deterministic transformation of a simpler base distribution $r$ such that $T_\lambda(z) \sim q_\lambda$ with $z \sim r$. This allows writing an expectation with respect to $q_\lambda$ as an expectation over the simpler distribution $r$. The reparameterization estimator corresponds to using $g(z_s) = \nabla_\lambda f(w(T_\lambda(z_s)))$ (for $z_s \sim r$) in place of $g(\theta)$, where $w$ implicitly depends on $\lambda$ as well. In the case of the adaptive $f$-divergence, the importance weights $w(\theta_1), \ldots, w(\theta_S)$ are sorted, and the gradients corresponding to each sample are then weighed by the empirical rank. The gradient estimates can be used in a stochastic gradient optimization scheme such that

$$\lambda^{t+1} \leftarrow \lambda^t + \eta_t \widehat{G}(\lambda^t), \tag{3}$$

where $\eta_t$ is the step size. In practice, more stable adaptive stochastic gradient optimisation methods such as RMSProp or Adam [9, 13], which smooth or normalize the noisy gradients, are often used.

Numerous prior work have studied some of the challenges tied to optimizing these divergence measures under the presence of noisy gradient estimates [1, 10, 29, 31]. Particularly, when dealing with mass-covering divergences, the gradient estimates can become so noisy that convergence is not possible in practice, as we will illustrate later on.

## 3 Assessing the Reliability of Black-box Variational Inference

### 3.1 Conceptual framework

How can we determine – both conceptually and experimentally – what is required to obtain reliable estimates of the variational divergence and optimal variational approximation? As we have seen, the most common variational divergences and their Monte Carlo gradient estimators can be expressed in terms of the density ratio $w(\theta)$. Reliable black-box variational inference ultimately depends on the behavior of $w(\theta)$ since (1) accurate optimization requires low-variance and (nearly) unbiased gradient estimates $\widehat{G}(\lambda)$, and (2) determining convergence and validating the quality of variational approximations can require accurate estimates $\widehat{L}(\lambda)$ of variational divergences [14, 15]. While *asymptotically* (in the number of iterations and Monte Carlo sample size $S$) there may be no issues with stochastic optimization or divergence estimation, in practice black-box variational inference operates in the *pre-asymptotic* regime. **Therefore, the reliability of black-box variational inference depends on the pre-asymptotic behavior of the $w(\theta)$, and how it interacts with the choice of variational objective and gradient estimator.**

Before accounting for the effects of the objective and gradient estimator, first consider the behavior of the density ratio $w(\theta)$, which can also be interpreted as an importance sampling weight with $q_\lambda(\theta)$ as the proposal distribution [cf. 2, 16, 29]. Pickands [23] proved, under commonly satisfied conditions, that for $u$ tending to infinity, the distribution of $w(\theta) \mid w(\theta) > u$ is well-approximated by the three-parameter generalized Pareto distribution $\mathsf{GPD}(u, \sigma, k)$, which for $k > 0$ has density $p(w \mid u, \sigma, k) = \sigma^{-1}\{1 + k(w - u)/\sigma\}^{-1-1/k}$ where $w$ is restricted to $(u, \infty)$. Since $w(\theta) > 0$, this implies its distribution is heavily skewed to the right with a power-law tail. Consider the idealized scenario of estimating the mean of $w(\theta) \sim \mathsf{GPD}(u, \sigma, k)$. We assume the mean is finite, which is equivalent to assuming $k < 1$ since $\lfloor 1/k \rfloor$ determines the number of finite moments. Because of the heavy right skew, most of the mass of $w(\theta)$ is below its mean. Therefore, even after averaging a large number of samples, most empirical estimates $\sum_{s=1}^S w(\theta_s)$ will be smaller than the true mean. Figure 3a illustrates this behavior for different values of $k$: even with 1 million samples, the empirical mean is far below the true mean when $k > 0.7$. The highly variable sizes of the confidence intervals based on 10,000 replications further highlight the instability of the estimator. **So, even though the empirical mean is an unbiased estimator, in the pre-asymptotic regime (before the generalized central limit theorem is applicable [5]), in practice the estimates are heavily biased downward with high probability.** If $w(\theta)$ is not a generalized Pareto distribution, we can instead treat $k$ as

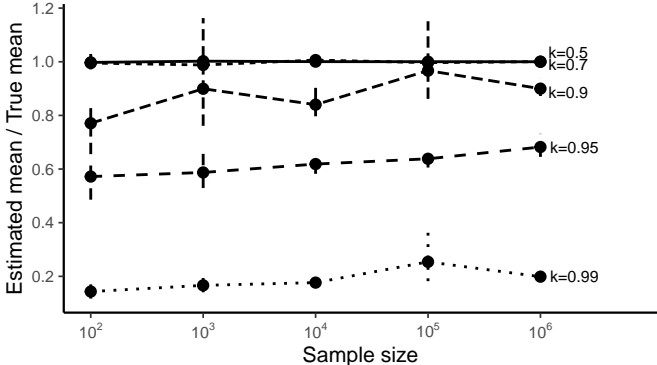

Figure 2: The ratio of estimated mean and true mean for different values of $k$ shape parameter of a generalized Pareto distribution and confidence intervals in a finite sample size simulation.

the *tail index* $k := \inf\{\ell > 0 : \mathbb{E}_{\theta \sim q}\{w(\theta)^{1/\ell}\} < \infty\}$, which encodes the same tail behavior as $\mathsf{GPD}(u, \sigma, k)$. Crucially, we should expect $k$ to be much larger than 0 when there is a significant mismatch between the target distribution and the variational family. **Since selecting a variational family that can match the typical set tends to be more difficult in higher dimensions, we should expect $k$ to be larger for higher-dimensional posteriors.**

We can generalize our observations about pre-asymptotic estimation bias to the estimators $\widehat{L}(\lambda)$ and $\widehat{G}(\lambda)$. For the loss estimator, we replace $w(\theta_s)$ with $f(w(\theta_s))$, where $f(w)$ is polynomial in $w$ and $\log w$ for the class of losses we consider. If the dominant term of $f(w)$ is of order $w^\alpha$, the tail behavior will be similar to a generalized Pareto with $k_\alpha = \alpha k$. Thus, $\widehat{L}(\lambda)$ will have larger pre-asymptotic bias as $\alpha$ increases. For example, estimation of the mass-covering inclusive KL (where $\alpha = 1$) – and, more generally, mass-covering $\alpha$-divergences with $\alpha > 0$ – will suffer from a large pre-asymptotic bias. On the other hand, for the mode-seeking exclusive KL, $f(w) = \log(w)$, so we can expect all moments to be finite and therefore a much smaller pre-asymptotic bias.

Similar considerations apply to the gradient estimator, with the details depending on the specific estimator used. However, when using self-normalized weights for $\alpha$-divergences, we can expect a large pre-asymptotic bias whenever $w(\theta)$ has such bias since self-normalization involves estimating the mean of $w(\theta)$. This bias will affect the accuracy of the solution found using stochastic optimization. Thus, the quality of the solutions found can only partially be improved by using a smaller step size since smaller step sizes will only reduce the effects of a large estimator variance, but not the effects from a large bias. We provide more details on the behavior of the score function and reparameterized gradients for each of the divergences in **????**, following Geffner and Domke [11].

In summary, our framework makes two key predictions:

**(P1)** Estimates and gradients of mode-seeking divergences (in particular exclusive KL divergence with log dependence on $w$) have lower variance and are less biased than those of mass-covering divergences (in particular $\alpha$-divergences with $\alpha > 0$, with polynomial dependence on $w$).

**(P2)** The degree of polynomial dependence on $w$ determines how rapidly the bias and variance will increase as approximation accuracy degrades – in particular, in high dimensions.

Because the adaptive $f$-divergence depends directly on the (ordered) weights, we expect it to behave similarly to the mass-covering divergences.

### 3.2 Experimental framework

In the light of potentially large non-asymptotic bias arising from the heavy right tail of $w(\theta)$, it is important to verify the pre-asymptotic behavior of the Monte Carlo estimators used in variational inference. We follow the approach developed by Vehtari et al. [27] for importance sampling and compute an empirical estimate $\hat{k}$ of the tail index $k$ by fitting a generalized Pareto distribution to the observed tail draws. In the importance sampling setting, Vehtari et al. [27] show that the minimal sample size to have a small error with high probability scales as $S = \mathcal{O}(\exp\{k/(1-k)^2\})$. Vehtari

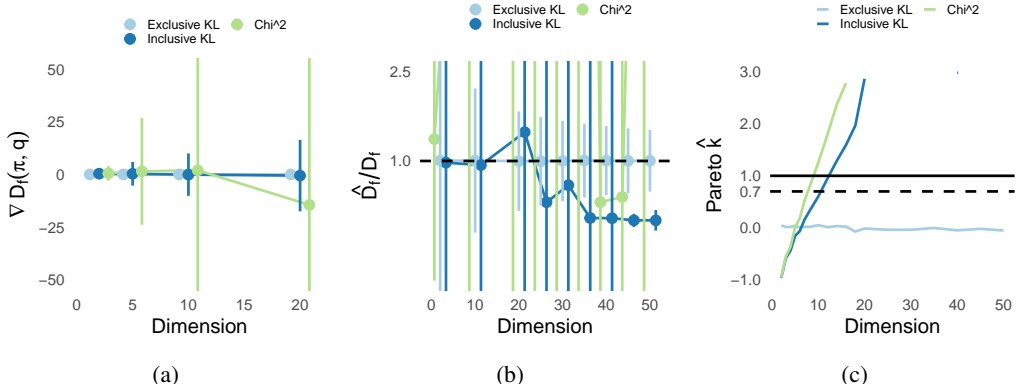

Figure 3: Results for correlated Gaussian targets of dimension $D = 1, \ldots, 50$ using either the exclusive or inclusive KL divergence as the variational objective. Each line in the plots corresponds to fitting and evaluating the same divergence measure as indicated in the legend. Each result is the average of 50 independent simulations. Quantiles are computed from simulating $100,000$ draws. **(a)** Bias and variance of the gradients of the optimised $f$-divergence for one parameter $\theta_d$ for increasing dimensions at the end of the optimisation for correlated Gaussian targets of dimension $D = 1, \ldots, 20$ and mean field Gaussian as variational approximation. **(b)** The ratio of the $f$-divergence estimate to the true value. **(c)** The $\hat{k}$ values for the variational approximations.

et al. [27] also demonstrate that $\hat{k}$ provides a practical *pre-asymptotic convergence rate estimate* even when the variance is infinite and a generalized central limit theorem holds. While estimating $\hat{k}$ in general requires larger sample size than is commonly used to estimate the stochastic gradients, we can still use it to diagnose and identify the challenges with different divergences. If $\hat{k} > 0.7$, the minimal sample size to obtain a reliable Monte Carlo estimate is so large that it is usually infeasible in practice. This cutoff is in agreement with our findings shown in Fig. 3a. Thus, together with our conceptual framework, we have a third key prediction:

**(P3)** The $\hat{k}$ value can be used to diagnose pre-asymptotic reliability of variational objectives. In particular, the $\alpha$-divergence with $\alpha > 0$ will become unreliable when $\max(1, \alpha) \times \hat{k} > 0.7$, even if $w$ is bounded (by a very large constant).

### 3.3 Verification of Pre-asymptotic (Un)reliability

We first verify our three key predictions in a simple setting where we can compute most of the relevant quantities such as the loss function in closed form. Specifically, we fit a mean-field Gaussian to a Gaussian with constant $0.5$ correlation factor using the inclusive KL, exclusive KL, $\chi^2$, and $1/2$-divergences. We vary the dimensionality $D$ from 1 to 50, which is a surrogate for the degree of mismatch between the optimal variational approximation and the target distribution. To find the optimal divergence-based approximation, we optimize the closed-form expression for the divergences between two Gaussians. Hence, we can consider on the *best-case scenario* and ignore the complexities and uncertainty due to the stochastic optimization. Due to space limitations, we focus on representative cases of the approximations from optimising the mode-seeking exclusive KL divergence and the mass-covering inclusive KL divergence. Results for the other divergences are included in the appendix.

**(P1) Mode-seeking divergences are more stable and reliable than mass-covering ones.** Figure 3b shows that as the approximation–target mismatch increases with dimension, the bias in and variance of the divergence estimates increases substantially for the inclusive KL and $\chi^2$ but only moderately for the exclusive KL. Similarly, Fig. 2 shows that gradient bias and variance increases with dimension for inclusive KL and $\chi^2$ but not exclusive KL.

**(P2) Degree of polynomial dependence on $w$ determines sensitivity to approximation–target mismatch.** Figure 3b shows that divergence estimates resulting from optimising higher polynomials of $w$ become more and more unstable as dimensions increases.

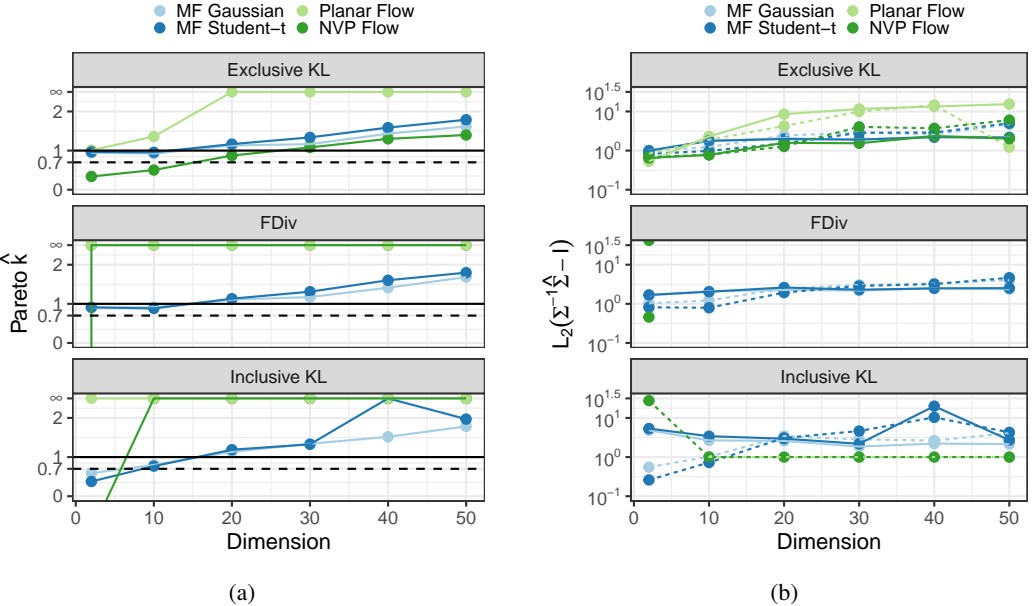

Figure 4: Results for increasing dimensions of the robust regression model. **(a)** Pareto $\hat{k}$ values for BBVI approximations. **(b)** Relative error of covariance estimates for BBVI (solid lines) and after PSIS correction (dashed lines).

**(P3)** $\hat{k}$ **diagnoses pre-asymptotic reliability.** Figure 3c shows that the $\hat{k}$ values grow rapidly for the inclusive KL-based approximation, particularly for higher-degree dependence on $w$, which agrees with predicted behavior and large bias and variance of the inclusive KL and $\chi^2$. In contrast, the $\hat{k}$ values remain fairly stable for the exclusive KL-based approximation, again in agreement with predicted and observed bias and variance behavior.

## 4 Experiments

In this section, we describe a series of experiments to study how our pre-asymptotic framework can be used for assessing the reliability of black-box variational approximations for practical applications and developing best-practices. For all posteriors, we fit mean-field Gaussian and Student-$t$ families, a planar flow [25] with 6 layers and a non-volume preserving (NVP) flow [8] with 6 stacked neural networks with 2 hidden layers of 10 neurons each for both the translation and scaling operations with a standard Gaussian distribution for the latent variables. We use Stan [26] for model construction. For stochastic optimization we use RMSProp with initial step size of $10^{-3}$ run for either $T_{\max}$ iterations or until convergence was detected using a modified version of the algorithm by Dhaka et al. [6]. For the exclusive KL we use 10 draws for gradient estimation per iteration, while for the other divergences we use 200 draws, and a warm start at the solution of the exclusive KL. In practice, we found the optimisation for $\chi^2$ divergence extremely challenging, with the solution failing to converge even for moderate dimensions $D \approx 10$. Therefore, we only include results for the KL divergences and the adaptive $f$-divergence. We compare the accuracy of approximated posterior moments to ground-truth computed either analytically or using the dynamic Hamiltonian Monte Carlo algorithm in Stan [26]. Specifically, we consider the estimates $\hat{\mu}$ and $\hat{\Sigma}$ for, respectively, the posterior mean $\mu$ and covariance matrix $\Sigma$. We also consider the mean and covariance estimates produced by PSIS and compute $\hat{k}$. The experiments were carried on a laptop and an internal cluster with only CPU capability. The code for the experiments will be made available after acceptance using MIT license.

### 4.1 Heavy-tailed posteriors

First, we study the toy robust regression model previously used by Huggins et al. [14] given by

$$\beta_d \sim \mathrm{N}(0, 10), \qquad\qquad y_n \mid x_n, \beta \sim t_{10}(\beta^\top x_n, 1),$$

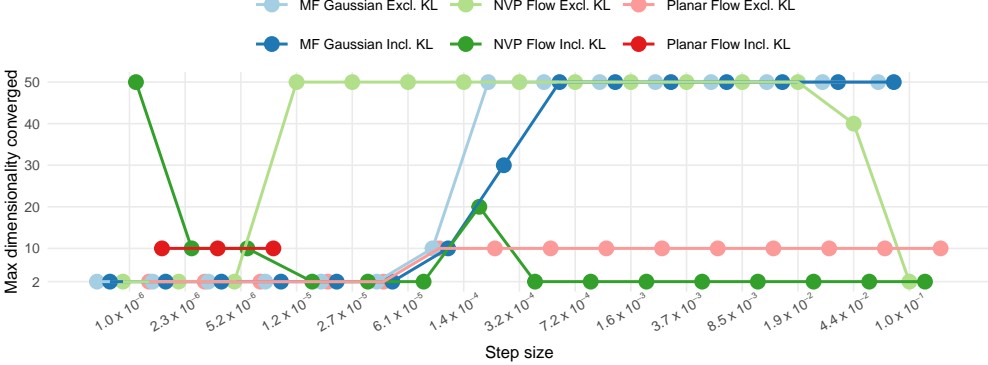

Figure 5: Maximum dimensionality converged per step size for the robust regression model.

where $y_n \in \mathbb{R}, x_n \in \mathbb{R}^D$ are the target and predictors respectively, $\beta$ denotes the unknown coefficients, and $D$ is varied from 2 to 50. We generated data from the same model with covariates generated from a zero-mean Gaussian with constant correlation of $0.4$. The Student's $t$ leads to the posterior having heavy tails, making it a more challenging target distribution. We use $T_{\max} = 10{,}000$.

**Mode-seeking divergences are easier to optimize.** Figure 4a shows that the estimated tail index $\hat{k}$ generally increases with the dimension as expected. In particular, the $\hat{k}$ values when using normalizing flows, which are more challenging to optimize, is low for $D < 20$ when using exclusive KL, but infinite when using either the inclusive KL or $f$-divergence. From Fig. 4b we can see that exclusive KL provides also more accurate and reliable posterior approximations than the inclusive KL and adaptive $f$-divergence, particularly for the normalizing flows. This observation is consistent with the prediction (P3) of the proposed framework. The better performance for normalizing flows corroborates the relative ease of stochastic optimization with the exclusive KL divergence compared to the inclusive KL or the adaptive $f$-divergence – despite the fact that we used 20 times as many Monte Carlo samples to estimate the gradients for the inclusive KL and the $f$-divergence compared to the exclusive KL. To further illustrate the relative difficulty of optimizing the inclusive KL divergence, Fig. 5 shows the largest dimension for which the stochastic optimization converged as a function of the step-size. For most step-sizes, the combination of normalizing flows and the inclusive KL divergence only converged for $D = 2$, whereas convergence is possible in higher dimensions for simpler variational families. These observations are consistent with predictions (P1)-(P2) of the proposed framework.

**Adaptive $f$-divergence interpolates between the exclusive and inclusive KL divergence, but is difficult to optimize.** In low dimensions, the adaptive $f$-divergence behaves somewhere between the two KL divergences as seen in **??** – as it was designed to [29]. As confirmed by Fig. 4, For higher-dimensional posteriors, we expect it to behave more like the exclusive KL, but it is less stable due to its functional dependence on the importance weights.

**Normalizing flows can be effective but are challenging to optimize.** Fig. 4 also shows that normalizing flows can be quite effective when used with exclusive KL to ensure stable optimization. However, as can be seen in **??**, when using out-of-the-box optimization with no problem-specific tuning (as we have done for a fair comparison), the normalizing flows approximations can have pathological features – even in low dimensions.

## 4.2 Realistic models and datasets

We now study how the choice of divergence and approximating family compare across a diverse range of benchmark posteriors. We compare variational approximations for models and datasets from posteriordb[*] in terms of accuracy of the estimated moments and predictive likelihood. We used an 80/20 training/test split on all datasets to compute the predictive likelihoods. We use $T_{\max} = 15{,}000$.

---

[*] https://github.com/stan-dev/posteriordb

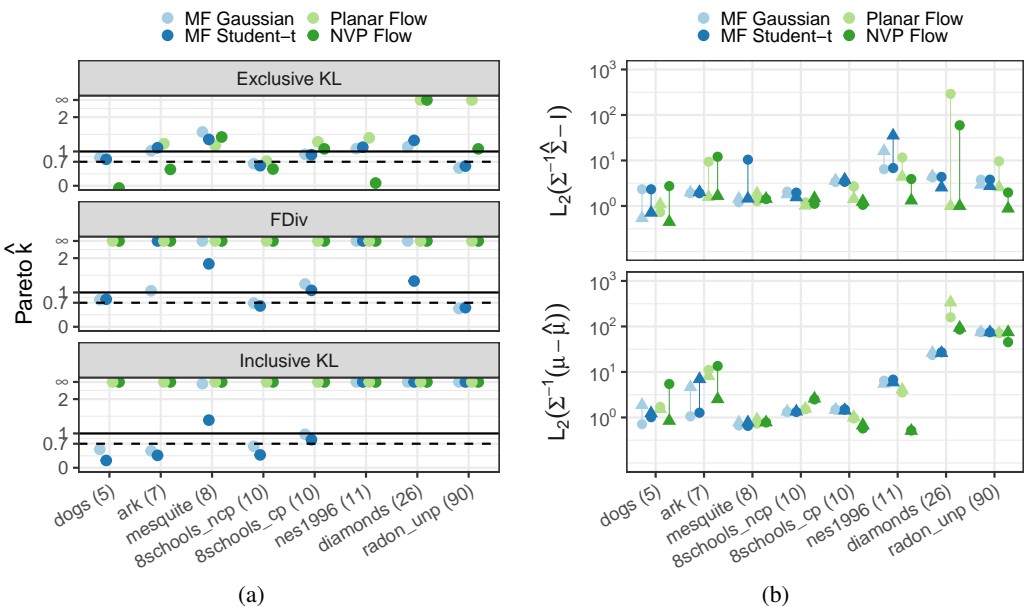

Figure 6: Results for `posteriordb` experiments. Dimensionality of each dataset is given in parentheses. **(a)** Pareto $\hat{k}$ values for BBVI approximations. **(b)** Relative error of mean and covariance estimates for BBVI using exclusive KL (circles) and after PSIS correction (triangles).

Table 1: Predictive likelihood results on `posteriordb` datasets using a Mean Field Gaussian approximation. The results denote the likelihood with the variational approximation solution obtained and after PSIS correction to the solution. **Bold** (underline) indicates best-performing method(s) (variational method(s))

| Name | HMC | Excl. KL | Excl. KL+PSIS | Incl. KL | Incl. KL+PSIS |
|---|---|---|---|---|---|
| dogs | -71.1±1.2 | -71.2 ±1.3 | -71.7±1.5 | -110±3.5 | **-70.5**±4.1 |
| arK | **-32.4**±0.6 | **-34.3**±0.7 | **-34.4**±0.7 | -35.2±0.8 | -34.9±0.8 |
| mesquite | **-1681**±127 | -2512±140 | -5418±186 | -∞ | -∞ |
| nes1996 | **-412.9**±1.7 | **-412.8**±1.7 | -427.9±1.8 | -2140.5±59.7 | -499.3±45.6 |
| diamonds | **22.1**±3.1 | -2.6±1.3 | 1.5±1.2 | -3196.6±57.9 | -3149±55.7 |
| radon | **-234.4**±1.1 | -353.0±19.3 | -325.0±19.5 | -377.4±1.9 | -370.5±2.2 |

**Exclusive KL remains the most reliable for realistic posteriors.** The results are summarized in Fig. 6, where the same pattern is seen: the exclusive KL is superior for higher-dimensional posteriors (e.g., $D > 10$) or when combined with normalizing flows, while inclusive KL is better for lower-dimensional posteriors. Despite the superior performance of the exclusive KL divergence, the large values for $\hat{k}$ indicate that fitting approximations based on normalizing flows remains a challenge in high dimensions. The performance for the adaptive $f$-divergence is comparable to the inclusive KL divergence. Table 1 shows that the exclusive KL divergence consistently outperforms the inclusive KL divergence in terms of predictive accuracy, but can be significantly worse than HMC.

**Importance sampling can substantially improve accuracy.** Focusing on exclusive KL, Fig. 6b shows the relative errors of the first two moments for the variational approximation (dots) and after correcting the estimates using PSIS (triangles). In some cases, the PSIS correction dramatically improved the accuracy of the normalizing flows.

**Reparameterization is an important tool for improving accuracy.** The 8-schools model is low-dimensional ($D = 10$), but the funnel-shaped posterior makes inference challenging for variational approximations [14, 31]. As has been noted previously in the literature, and is clear from Figs. 6a and 6b, reparameterizing the model so that the posterior better matches the variational family can be an effective way to improve the accuracy of the approximation. See **??** for an illustration.

# 5 Discussion

Our conceptual framework based on the pre-asymptotic behavior of the density ratios / importance weights $w$ along with our comprehensive experiments lead to a number of important takeaways for practitioners looking to obtain reasonably accurate posterior approximations using black-box variational inference:

- The instability of mass-covering divergences like inclusive KL and $\chi^2$ means that, given currently available methodology, users are better off using the exclusive KL divergence except for easy low-dimensional posteriors. The reliance of the adaptive $f$-divergence on importance weights leads to similar instability.

- Importance sampling appears to almost always be beneficial for improving accuracy, even when the $\hat{k}$ diagnostic is large. However, a large $\hat{k}$ does suggest the user should not expect even the PSIS-corrected estimates to be particularly accurate.

- Using normalizing flows – particularly NVP flows – together with exclusive KL and PSIS provides the best and most consistent performance across posteriors of varying dimensionality and difficulty. We therefore suggest this combination as a good default choice.

Our results suggest an important direction for future work is improving the stability of optimization with normalizing flows, which still tend to have some pathological behaviors unless they are very carefully tuned since such tuning significantly detracts from the benefits of using BBVI.

# 6 Limitations

While our experiments included a range of common statistical model types, our findings may not generalize to all types of posteriors or to other variational families. For example, we did not explore semi-implicit methods or applications to neural networks. We also did not investigate alternative divergences such as those used in importance-weighted autoencoders.

### Acknowledgments

We thank Sihan Liu for pointing out and fixing errors in the code. We thank Ari Hartikainen and PyStan developers for help with posteriordb, Academy of Finland (grants 298742, and 313122) and Finnish Center for Artificial Intelligence for partial support of the research. We also acknowledge the computational resources provided by the Aalto Science-IT project. AKD would like to thank Tomas Geffner and Eero Siivola for useful discussions.

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
