Table A.1: Datasets from PosteriorDB.

| Name | Dimensions |
|---|---|
| Dogs | 5 |
| Ark | 7 |
| Mesquite | 8 |
| Eight schools non centered | 10 |
| Eight schools centered | 10 |
| NES1996 | 11 |
| Diamonds | 26 |
| Radon unpooled | 90 |

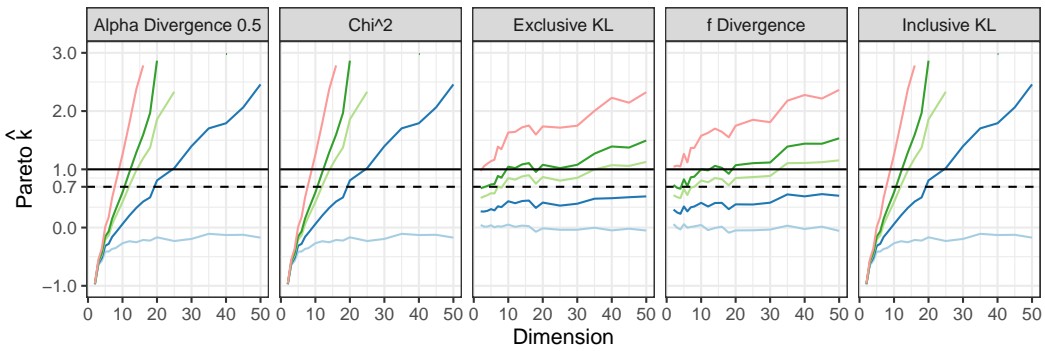

Figure B.1: Pareto $\hat{k}$ estimated for different objectives and divergences estimation for a $0.5$ correlated Gaussian target and mean field Gaussian approximation and increasing dimensionality. Here we compute the $\hat{k}$ for all the f(w) after optimizing a particular variational objective.

## A  PosteriorDB datasets

In Table A.1 we show the dimensionality of the datasets we use for our real experiments.

## B  Additional results for the pre-asymptotic reliability case study

In Fig. B.1 and Fig. B.2 we show additional results for the pre-asymptotic reliability case study for different objectives and mean field Gaussian approximation. The results from optimising $\chi^2$, $1/2$-divergence and tail adaptive $f$-divergence follow similar trends as those resulting from optimising exclusive and inclusive KL. Approximations obtained by optimising $\chi^2$ and $1/2$-divergence are more unstable and end up diverging in similar ways as inclusive KL even for moderately low dimensional problems. We use a warm start procedure for $\chi^2$, $1/2$-divergence and inclusive KL, starting at the solution of exclusive KL for a given problem. On the other hand, optimising tail adaptive $f$-divergence seems to be more robust and behave similarly to exclusive KL even in higher dimensions.

## C  Additional experiments

**Isolating the effect of variational family.** In this section, we perform a systematic comparison of inclusive-KL and exclusive KL divergences using mean-field Gaussian and mean-field Student-$t$ approximation families for varying amount of correlation and dimensionality of the underlying parameter space. The dimension size is varied from 2 to 100. For Gaussian target with Gaussian approximation, we used BFGS as the optimiser removing any error due to stochastic optimisation. The plots in Fig. C.3 show how $\hat{k}$ behaves with increasing dimension and increasing correlation in posterior for a mean field Gaussian approximation and optimising objectives for exclusive KL and inclusive-KL

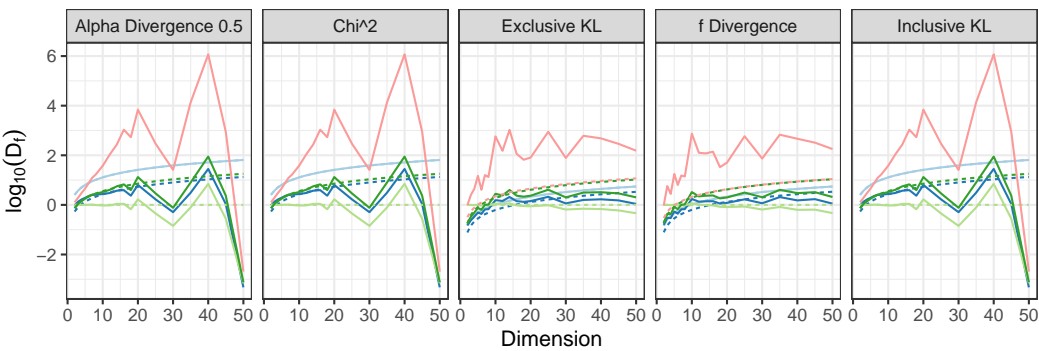

Figure B.2: Divergences estimates for different objectives for a $0.5$ correlated Gaussian target and mean field Gaussian approximation and increasing dimensionality.

divergences respectively. We also plot a similar plot for planar-flow when optimising exclusive KL divergence. The plots indicate even when the approximation is heavy tailed and divergence measure is *mass covering*, the final variational mean field approximation becomes unreliable. The dimension at which this happens depends on the posterior geometry (correlation in this case). Since the target is Gaussian, when the approximation is Gaussian family, we can estimate exclusive and inclusive KL analytically at the optimisation end points for each of the divergences, for the other approximations, Student's $t$ and planar flow, we estimate these quantities by MC.

Extensive experiments and results are shown in Fig. C.4, Fig. C.5, Fig. C.6 and Fig. C.7

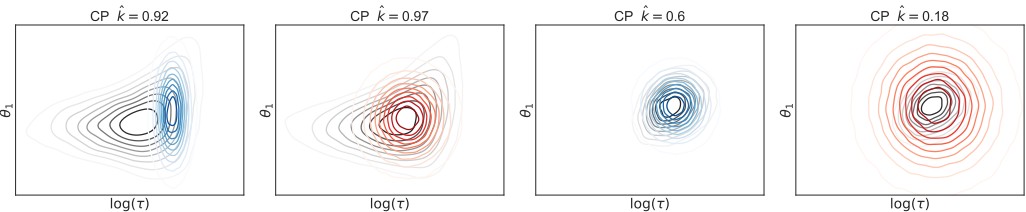

(a) results with centered parameterisation(CP) on standard eight schools and eight schools with more informative data.

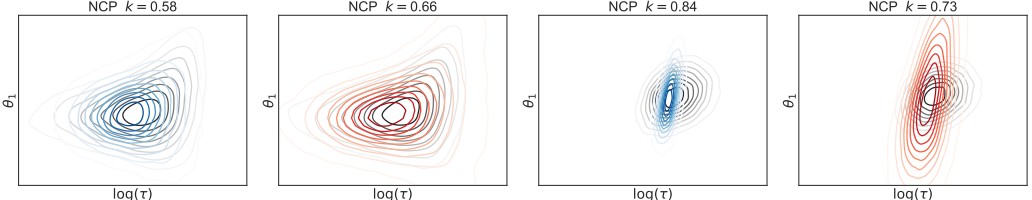

(b) results with non-centered parameterisation(NCP) on standard eight schools and eight schools with more informative data.

Figure C.1: Plots for the approximate posteriors obtained by optimizing exclusive KL(blue) and inclusive KL(red).

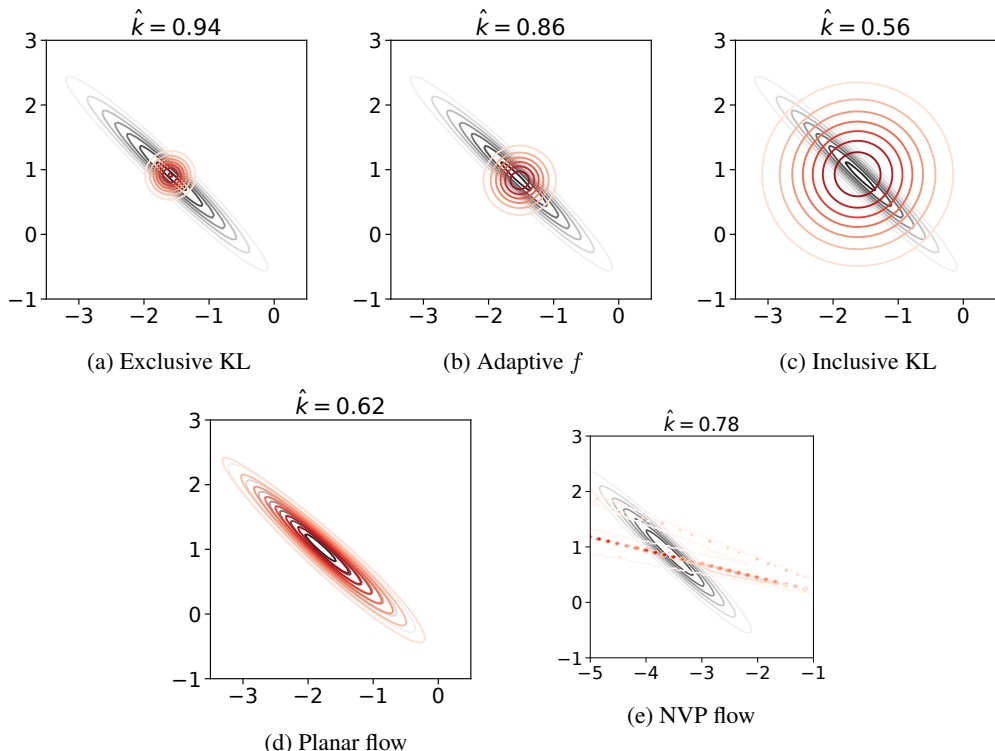

Figure C.2: Variational approximations (red) for robust regression posterior (black) with $D = 2$. **(a–c)** Uses mean-field Gaussian family. **(d,e)** Uses exclusive KL divergence.

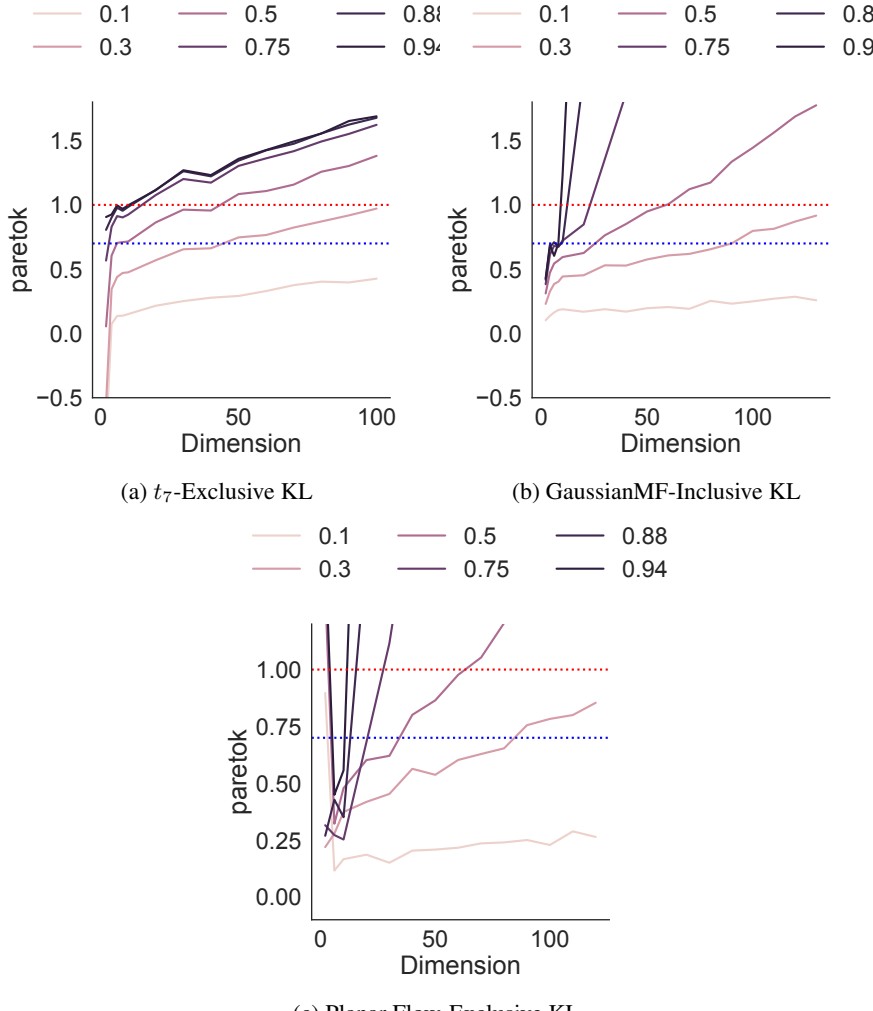

(a) $t_7$-Exclusive KL

(b) GaussianMF-Inclusive KL

(c) Planar Flow-Exclusive KL

Figure C.3: Plots of $\hat{k}$ with Exclusive KL divergence minimisation, Inclusive KL divergence minimisation with mean-field Student-$t$ density and with Planar Flows for increasing correlation and dimensions.

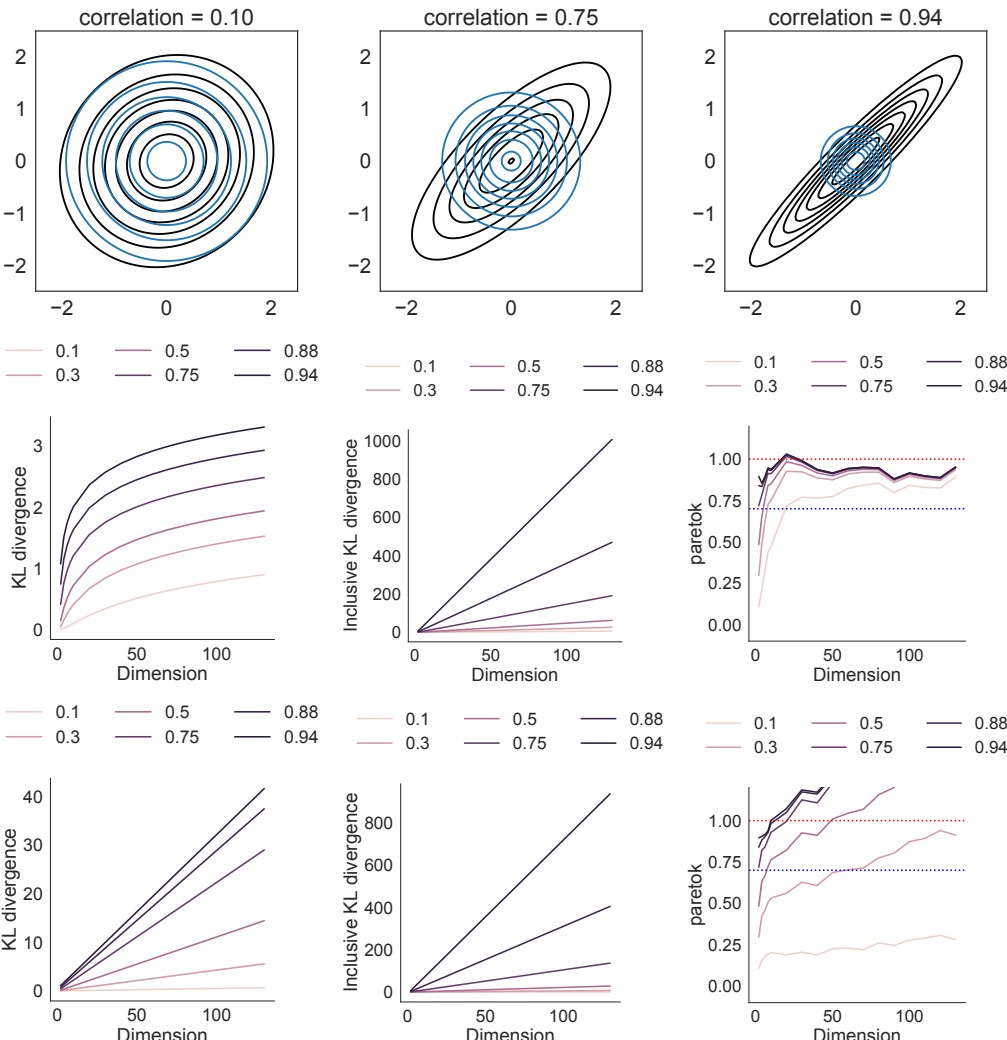

Figure C.4: Gaussian mean field solution for exclusive KL divergence. The top row shows solutions obtained after minimizing the exclusive KL divergence where the target is a correlated Gaussian density with varying amount of correlations, and the approximation is a mean field approximation. The second row shows plots from the left to the right: the exclusive KL divergence, the inclusive KL divergence and the Pareto $k$ statistic computed at the solution returned by BFGS optimisation for increasing dimensions and different amount of correlations when the target has a uniform covariance structure, the bottom row shows the corresponding plots for banded covariance target.

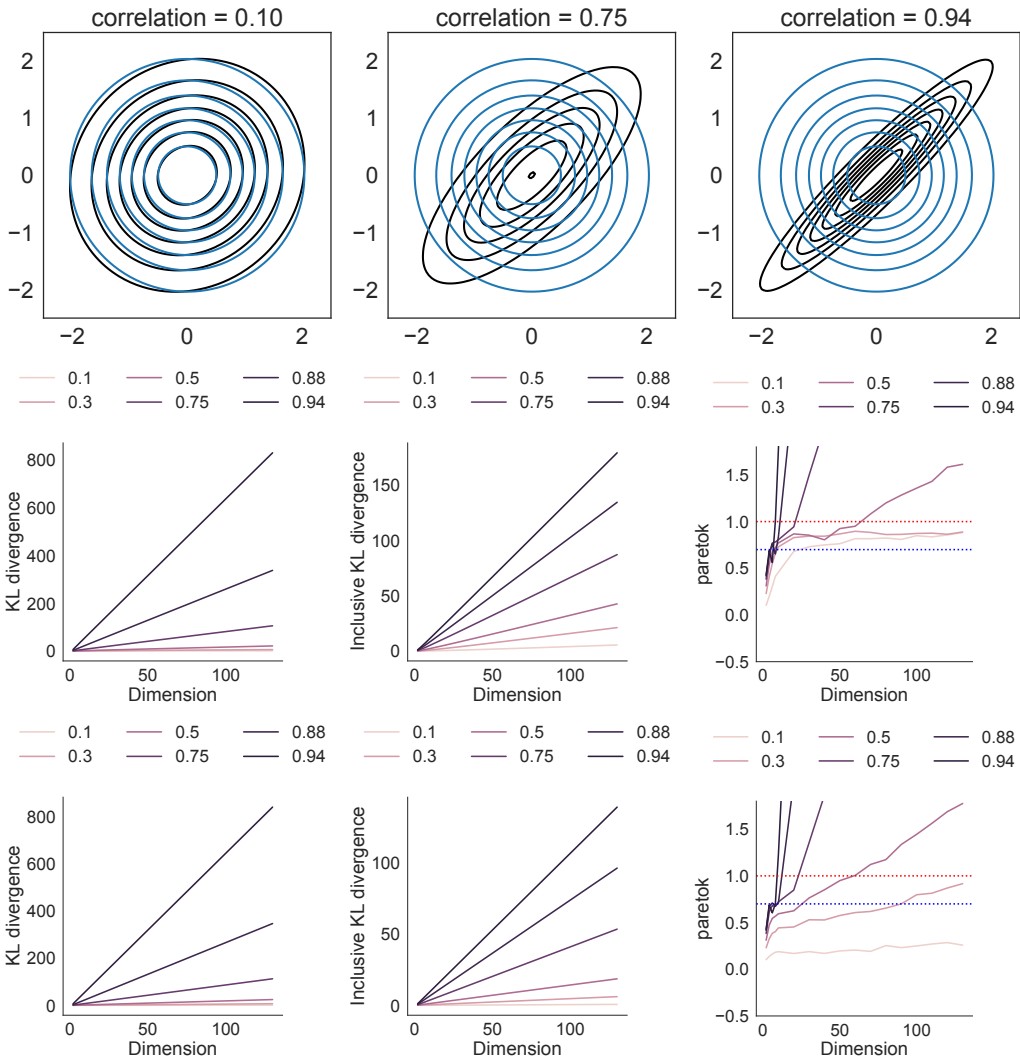

Figure C.5: Gaussian mean field solution for inclusive KL divergence. The top row shows solutions obtained after minimizing the exclusive KL divergence where the target is a correlated Gaussian density with varying amount of correlations, and the approximation is a mean field approximation. The second row shows plots from the left to the right: the exclusive KL divergence, the inclusive KL divergence and the Pareto $k$ statistic computed at the solution returned by BFGS optimisation for increasing dimensions and different amount of correlations when the target has a uniform covariance structure, the bottom row shows the same corresponding plots for banded covariance target.

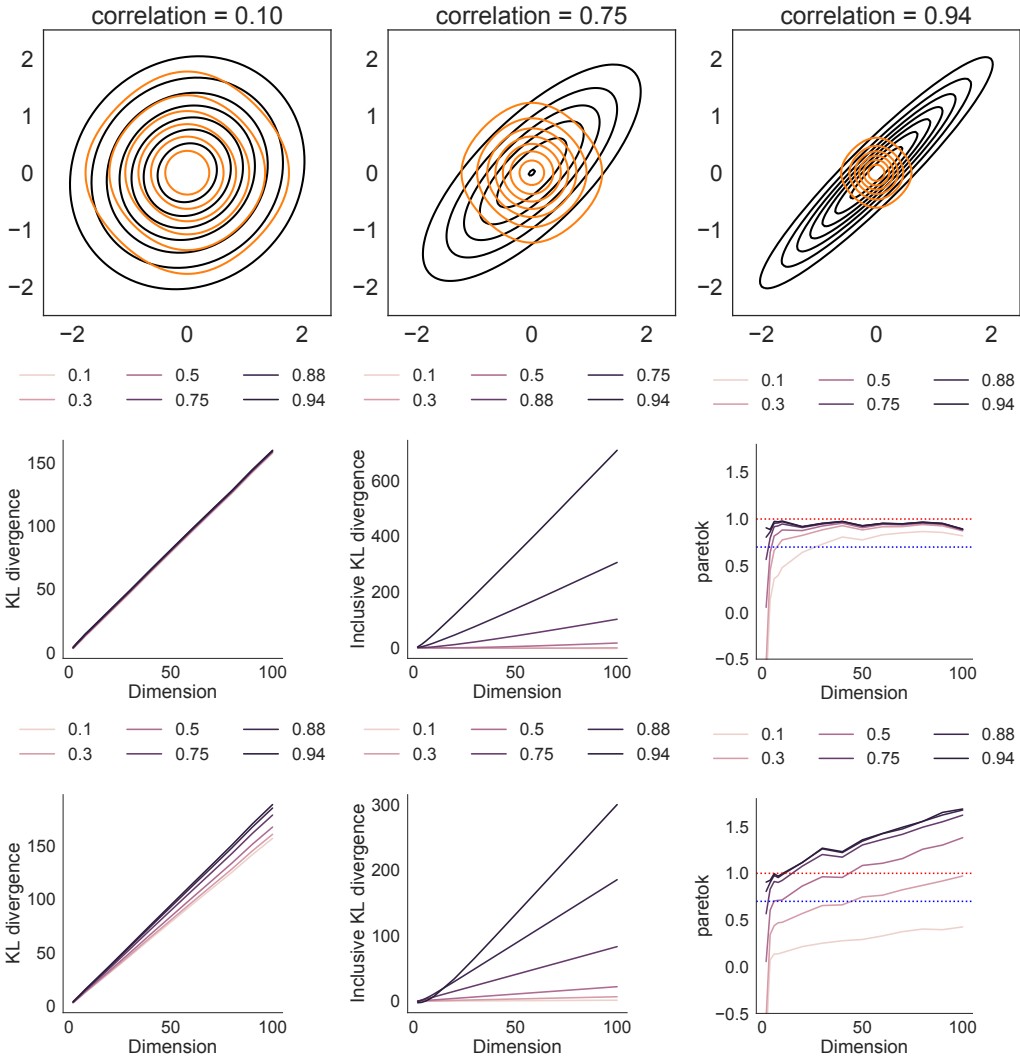

Figure C.6: Solution for exclusive KL divergence where the family of approximation is a product of t-densities. The top row shows solutions obtained after minimizing the exclusive KL divergence where the target is a correlated Gaussian density with varying amount of correlations, and the approximation is a mean field approximation. The second row shows plots from the left to the right: the exclusive KL divergence, the inclusive KL divergence and the Pareto $k$ statistic computed at the solution returned by stochastic optimisation for increasing dimensions and different amount of correlations when the target has a uniform covariance structure, the bottom row shows the same corresponding plots for banded covariance target.

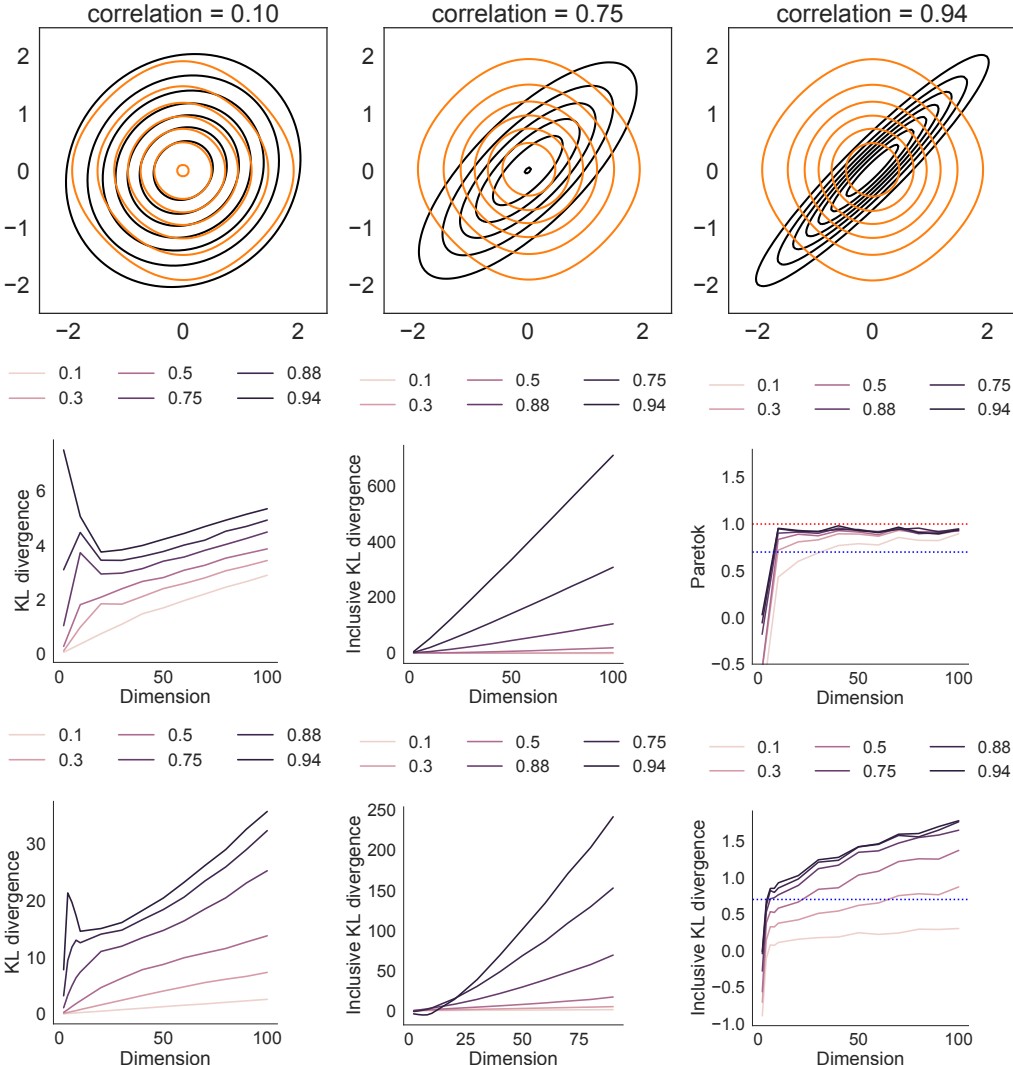

Figure C.7: Solution for inclusive KL divergence where the family of approximation is a product of t-densities. The top row shows solutions obtained after minimizing the inclusive KL divergence where the target is a correlated Gaussian density with varying amount of correlations, and the approximation is a mean field approximation. The second row shows plots from the left to the right: the exclusive KL divergence, the inclusive KL divergence and the Pareto $k$ statistic computed at the solution returned by stochastic optimisation for increasing dimensions and different amount of correlations when the target has a uniform covariance structure, the bottom row shows the same corresponding plots for banded covariance target.

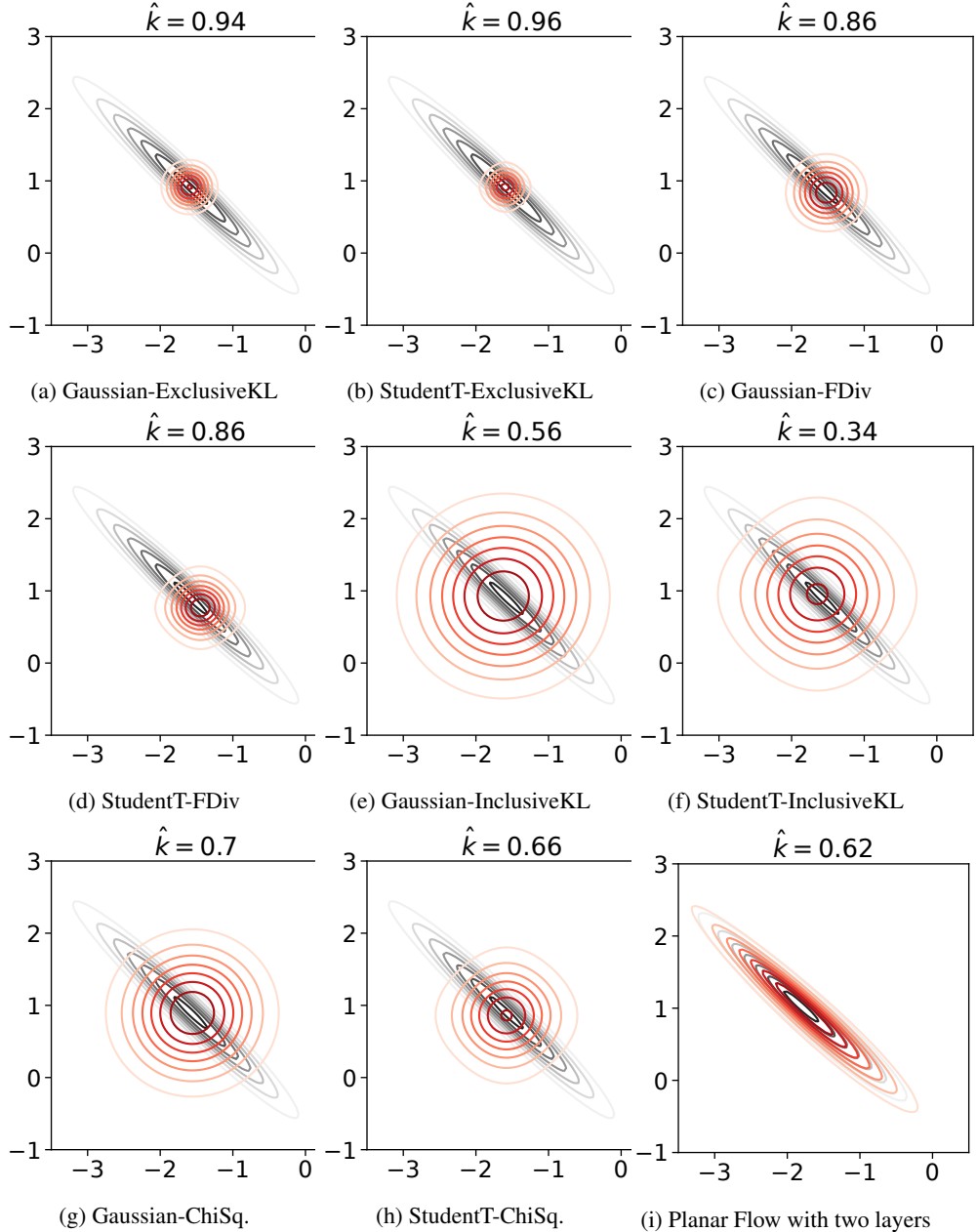

Figure C.8: Approximation for Robust Regression with different divergences and approximation families in 2 dimensions. This shows the properties of divergences and approximations in low dimensions.

 **D   Score function additional discussion**

 The score function gradient for exclusive KL is given as:

$$\nabla_\lambda \mathrm{L}(\lambda) = \nabla_\lambda \mathrm{E}_q[\log p(Y,\theta) - \log q(\theta)]$$
$$= \mathrm{E}_{q_\lambda(\theta)}[\log p(Y,\theta) - \log q(\theta)] \cdot \nabla_\lambda \log q(\theta)$$
$$\approx \frac{1}{S}\sum_{s=1}^{S}[\log w_s \nabla_\lambda \log q_\lambda(\theta_s)],$$

where we have defined $w_s = w(\theta_s)$. If the entropy of the approximate distribution is known analytically, we get another unbiased gradient estimator, where we use the MC samples only to estimate the first part, removing any direct dependence of gradient wrt $w(\theta_s)$

$$\nabla_\lambda \hat{\mathrm{L}}(\lambda) = \frac{1}{S}\sum_{s=1}^{S}[\log p(Y,\theta_s)\nabla_\lambda \log q_\lambda(\theta_s)] + \nabla_\lambda \mathrm{H}_q[q_\lambda(\theta_s)].$$

For inclusive KL divergence, the score function gradient is given as:

$$\nabla_\lambda \mathrm{L}(\lambda) = -\sum_{s=1}^{S}\frac{w_s}{\sum_{s=1}^{S}w_s}\nabla_\lambda \log q_\lambda(\theta_s). \tag{D.1}$$

where the gradient has been estimated by self-normalised importance sampling [4, 16**?** ].

Similarly, the score gradient for $\chi^2$ and $\alpha$ divergences is given as

$$\nabla_\lambda \mathrm{L}(\lambda) = \frac{-1}{S}\sum_{s=1}^{S}[w_s^{\text{score}}]^\alpha \nabla_\lambda \log q(\theta_s;\lambda),$$

where $\alpha \geq 2$

It is apparent immediately that the gradients will have even higher variance than observed in the case of importance sampling. Importance sampling is known not to work well in higher dimensions, since the variance of the importance weights is likely to become very large or infinite.

The variance of the score gradients for the divergences discussed above as a function of density ratios is given below:

$$\mathrm{V}_q(G_{\text{CUBO}}^{\text{score}}) = O(w^4),$$
$$\mathrm{V}_q(G_{\text{Inclusive KL}}^{\text{score}}) = O(w^2),$$
$$\mathrm{V}_q(G_{\text{ExclusiveKL}}^{\text{score}}) = O(\log(w)^2).$$

The higher the power on density ratio, the faster the variance of the gradients will grow. This means the density ratio should have finite higher moments for CLT to apply as discussed in Section 2.

**E   Reparameterised gradients additional discussion**

For exclusive KL, the reparameterised gradient becomes

$$\nabla_\lambda \mathrm{E}_q[\log j(\theta)] = \mathrm{E}_p[\nabla_\lambda T_\lambda(\epsilon)\nabla_\theta \log j(\theta)]. \tag{E.1}$$

In the case of $\chi^2$ divergence, the reparameterised gradient is

$$\nabla_\lambda \hat{\mathrm{L}}(\lambda) = \frac{2}{S}\sum_{s=1}^{S}\left(\frac{j(T_\lambda(\epsilon_s))}{q(T_\lambda(\epsilon_s))}\right)^2 \nabla_\lambda \log\left(\frac{j(T_\lambda(\epsilon_s))}{q(T_\lambda(\epsilon_s))}\right),$$

which can be expressed in terms of density ratios as follows:

$$\nabla_\lambda \hat{\mathrm{L}}(\lambda) = \frac{2}{S}\sum_{s=1}^{S}\left(w_s^{\text{RP}}\right)^2 \nabla_\lambda \log\left(w_s^{\text{RP}}\right), \tag{E.2}$$

where the new weights $w^{\text{RP}}$ denote that they have been evaluated on samples obtained using the reparameterisation trick. In this case, the dependence of the gradient is not straightforward and also depends on the the product of the density ratio squared and its corresponding gradient.

## F   Covariance Structures

In this work, we use two types of covariance matrices, uniform matrices denoted by $U$: $K_{ij} = 1.[i = j] + \rho[i \neq j]$ and the banded structure, denoted by $B$: $K_{ij} = 1.[i = j] + \rho^{|i-j|}[i \neq j]$

## G   Gradient Variances for Score function gradient and RP gradient

We want to see how the variance of the gradients for different divergence objectives varies by extending the analysis from [29] Let us consider the log joint density cost function i.e $j(\theta) = \log p(Y, \theta) = \theta^2$ and $q(\theta) = \mathcal{N}(\mu, 1)$

Then for exclusive KL divergence, the RP gradient estimator is:

$$G^{\text{RP}} = \nabla_\lambda \mathbb{E}_q[j(\theta)] \tag{G.1}$$

$$G^{\text{RP}} = \mathbb{E}_\epsilon[\Delta_\mu^{\text{RP}}] = \mathbb{E}_\epsilon[\nabla_\mu T(\epsilon; \lambda) \nabla_\theta j(\theta)] \tag{G.2}$$

$$\Delta_\mu^{\text{RP}}(\text{KL}(q||p)) = 1.(2\theta) = 2(\mu + \epsilon) \tag{G.3}$$

$$\mathbb{V}(\Delta_\mu^{\text{RP}}(\epsilon; \lambda)(\mu))(\text{KL}(q||p)) = 4 \tag{G.4}$$

Since the gradient wrt the location parameter is a r.v, we can compute the variance under the standard distribution $N(0, 1)$. Similarly we can derive the variance of the score function gradient

$$G^{\text{score}}(\lambda) = \nabla_\lambda \mathbb{E}_q[j(\theta)] = \mathbb{E}_q[j(\theta) \nabla_\lambda \log q(\theta; \lambda)] \tag{G.5}$$

$$G^{\text{score}}(\lambda) = \mathbb{E}_q[\Delta_\mu^{\text{score}}] \tag{G.6}$$

$$\Delta_\mu^{\text{score}}(\text{KL}(q||p)) = \theta^2(\theta - \mu) \tag{G.7}$$

$$\mathbb{V}_q(\Delta_\mu^{\text{score}}(\theta; \lambda)(\mu))(\text{KL}(q||p)) = \mu^4 + 14\mu^2 + 15 \tag{G.8}$$

Now consider the score gradient for Inclusive KL and $\alpha$ divergences:

$$G_\alpha^{\text{score}}(\lambda) = \mathbb{E}\left[\frac{p(Y, \theta)}{q(\theta)}^\alpha \nabla_\lambda \log q(\theta_s; \lambda)\right] \tag{G.9}$$

Taking the target density, $p(Y, \theta) = -\theta^2/2$, where the factor $1/2$ helps in cancelling some terms. For the special case, $q(\theta) = N(\mu, 1)$, when $\mu = 0$, the two densities become equal and we are left only with $G_\alpha^{\text{score}}(\lambda) = \theta$. Then , but for a general case this is given as:

$$G_\alpha^{\text{score}}(\lambda) = \mathbb{E}\left[\frac{\exp(-\theta^2/2)}{\exp(-(\theta - \mu)^2/2)} \nabla_\lambda \log q(\theta_s; \lambda)\right] \tag{G.10}$$

$$= \mathbb{E}_q[\exp(\mu^2/2 + \theta\mu)^\alpha \nabla_\lambda \log q(\theta_s; \lambda)] \tag{G.11}$$

For the special case when $\alpha \to 1$, we get

$$G_\alpha^{\text{score}}(\lambda) = \exp(\mu^2/2)\mathbb{E}_q[\exp(\theta\mu)(\theta - \mu)] \tag{G.12}$$

$$\mathbb{V}(\Delta_\mu^{\text{score}}) = \exp(\mu^4/4)\mathbb{V}_q[\exp(\theta\mu)(\theta - \mu)] \tag{G.13}$$

when $\mu = 0$, meaning that the approximation is same as the target density, this reduces to $\mathbb{V}_q(\Delta_\mu^{\text{score}}(\theta; \lambda)(\mu)) = 1$(a constant), equal to the variance of a standard normal distribution.