# OpenReview forum: "Challenges and Opportunities in High Dimensional Variational Inference"
_NeurIPS.cc/2021/Conference — NeurIPS 2021 Poster_

### Official Review · Reviewer_c7n8 · 2021-07-10

**Rating:** 6
**Confidence:** 4

**Summary:**

This paper provides a statistical tool to analyze different variational inference methods in the f-divergence family. It showed that for high dimensional and low dimensional problems, the behavior of the variational inference methods may differ, and the analysis based on the GPD statistics can suggestion different strategies for different situations. The predictions from this theory were validated by some commonly used models (e.g., linear regression). In summary, this paper can provide useful guidelines for variational inference practitioners.

**Limitations And Societal Impact:**

Here are some more detailed commons of the paper:

1. The three predictions (P1, P2, P3) are reasonable and are verified by some experiments. However, more rigorous proofs or justifications are needed to generalize the conclusions to other applications scenarios. For example, the authors mentioned that application to neural networks is not covered theoretically or empirically.

2. It would be better to provide a table to summarize the conclusions in the paper, for low dimensional and high dimensional problems. Properties and behaviors of different variational inference method could be included in the table, together with the recommended inference method in the f-divergence family.

3. A minor technical issue, in line 103, for the normalized importance weight, there should be a 1/S term in the denominator.

4. A problem in notation: D is used for divergence, but also used for dimension.

**Main Review:**

The main novelty of this paper is to use GPD to model the density ratio, which reveals important properties of different variational inference methods in the f-divergence family, which includes the inclusive KL and exclusive KL approaches. This provides a practical tool for people to use for variational inference diagnosis. One particular conclusion from this analysis is that, for moderate-to-high-dimensional posteriors, exclusive KL divergence is a better choice than inclusive KL divergence. These conclusion can be very helpful for people to choose the right variational inference method in different scenarios.

**Time Spent Reviewing:**

4 hours

---

> ### Author Response · Authors · 2021-08-10
> **Reply to Reviewer**
>
> Thank you for your suggestions and feedback.
>
> C1. We agree with the reviewer that further work in this direction is definitely needed in order to establish more strict theoretical criteria. We have considered many diverse models, like auto regressive models (arK), hierarchical Gaussian (radon) and non-Gaussian(8-schools) models, logistic regressions(dogs) which should be sufficient for the scope of this work, a study for neural networks is probably better suited for the longer version of this work.
>
> C2. This is an excellent idea and we will include such a table in the camera ready version.
>
> C3 and C4: Thanks for pointing these out, we will fix these issues in the camera ready version.

---

### Official Review · Reviewer_x3v3 · 2021-07-12

**Rating:** 6
**Confidence:** 3

**Summary:**

In this paper, the authors study the pre-asymptotic behavior of the density ratios between the joint distribution and the variational approximation. The authors use generalized Pareto distribution to model this density ratio in the pre-asymptotic regime and make three major predictions for how BBVI methods should behave. They verify their predictions through extensive experimentation, and finally, provide actionable advice to BBVI practitioners (use flows with exclusive KL and PSIS as a good default.)

**Limitations And Societal Impact:**

Yes

**Main Review:**

I appreciate the authors for taking up this important direction in BBVI research. In the recent years, the number of components in BBVI methods have increased dramatically and it is important that we, as a community, analyze the existing methods. This paper takes some very concrete steps towards it.

Overall, I like the paper. It is clearly written for most part. However, I have a few concerns and seek explanations.

- Figure 6. I am not sure how to make sense of $\hat k$ when using flows. Even with exclusive KL, the $\hat k$ value reaches $\infty$ for moderate dimensions (less than 100). Still, the relative errors seem to suggest they work better than the other methods. How reliable is $\hat k$ when comparing different methods while keeping the variational objective same? Since $\hat k$ value for flows seems to be $\infty$ for moderately large models, the use of $\hat k$ as a diagnosis tool is rather limited?
 - Figure 4b. I don't see the Planar flow and NVP flow lines (Green lines) for Fdiv and Inclusive KL? Were they not calculated because the $\hat k$ is $\infty$? If so, I believe something is not consistent. There is Relative error estimates for NVP Flow even though the $\hat k$ estimate is $\infty$ for exclusive KL and D = 50. Maybe I missed the explanation somewhere; it will be great if the authors can point it out or provide one.
 - Table 1. I am not entirely sure the bold and the underline are supposed to be. Also, is the entire table only for NVP based method? I could not find, in text or in the table caption, the family that was used for the reported numbers. Are there any error bars for these numbers?
 - Figure 5. Should not there be at least on entry corresponding to the planar and NVP flows for higher dimensions (>30)? Did none of these step sizes converged for flows? If so, how did we get the numbers in Figure 4?
 - At L69, what is $\hat I$? an unbiased or biased estimator? This is more of a notational consistency issue.

**Time Spent Reviewing:**

3-4 hours

---

> ### Author Response · Authors · 2021-08-10
> **Reply to Reviewer**
>
> We thank the reviewer for raising important questions and positive review.
>
> C1: This is a good question, we will add the discussion to the main text. $\hat{k}$ is scale invariant, depending only on the distribution of important weights. While relative mean and covariance errors are good summary statistics for mean field gaussian posteriors, it does not tell the full story in more complex posteriors. In practice, this means that for posteriors other than mean field Gaussians, relative mean and covariance errors are looking at specific summaries of the distribution, which may not be the best indicator of overall accuracy, as indicated by $\hat{k}$. An example of this can be seen in Fig. C.2(e)
>
> C2: The issue for inclusive KL and F-div is that some of these models did not even converge because of numerical instabilities(and resulting divergences), and therefore we could not compute summary errors for any of them. We will update the text and clarify whether a model is simply not converging or if the approximation is very bad, but it converged. As we point out in the paper, exclusive KL tends to be more stable, even though in some cases the approximation is not great. We also point out that we attribute all $\hat{k}$ to infinite when they are over 2.1 with/without convergence issues(as recommended in [26] and [30]), which we will also clarify in the text. The computation budget was limited to 15000 iterations. We will distinguish the two pathologies by an asterisk(where convergence was not achieved) in the final draft.
>
> C3:Thank you for pointing out this omission. Table 1 refers only to mean field Gaussians approximations. Bold indicates the best overall method (including HMC), whereas underline shows the best variational result for each experiment. We will clarify both points in the caption.
>
> C4: The results we show in figure 4 are computed with a slightly different step size, which explain the differences with figure 5 results. We have updated all results to show in figure 4 results computed with a step size included in figure 5, so both figures are relatable. Note that even if $\hat{k}$ is very large, we can still compute some error summaries for some methods.
>
> C5: Yes the SNIS is a biased estimator, we will clarify this in the text.

---

### Official Review · Reviewer_EXRc · 2021-07-17

**Rating:** 7
**Confidence:** 3

**Summary:**

The paper discusses the accuracy of variational inference and the choices that need to be me made with regard to divergence and approximating family, particularly as they relate to the dimensionality of the problem. Based on an analysis of the pre-asymptotic behavior of the density ratios involved, the authors propose a framework based on Pareto smoothed importance sampling (PSIS) to help practitioners analyze the success of their approximate inference procedure. The authors provide experiments demonstrating general findings for variational inference for a range of divergences and approximate posteriors.

**Limitations And Societal Impact:**

The authors addressed some limitations of their work in that the discussion is limited to a typical f-divergences used in the literature and does not discuss all types of posteriors seen in the literature, such as semi-implicit posteriors. While not an extensive discussion, I found this description of limitations to be adequate for the goals of the work. I am not aware of potential negative societal impacts for this work.

**Main Review:**

Overall, I found the paper to be quite clear and well-written. I believe the paper to be quite significant as I am not aware of much work on trying to provide a framework for understanding variational inference and the choices that need to be made, particularly in a way that could be used by a practitioner with relative ease. I believe the insights on dimensionality, mode-seeking vs mode-covering, and the differences in the quality of approximation from different approximate posteriors (e.g. MF Gaussian, Student's t, NVP), are novel, valuable and provide good guidance for practitioners and those who regularly use VI in their research.

The main issue I had with the paper was the lack of a proper review of prior work. The paper does reference some prior work in passing, but it does not really discuss the prior work (for instance [14, 30]) in the detail that I think would be required. For instance, [30] quite similarly proposes using the $\hat{k}$ obtained from PSIS to evaluate how successful VI is - although they did not analyze different divergences from the standard mode-seeking one nor the effect of dimensionality. I think the paper would really require a proper discussion of these papers, as well as other attempts to assess the quality of VI approximations.

Overall, as someone who focuses on VI in their research, I think this is quite a good paper and represents a useful and needed direction for the field. Therefore I believe the paper deserves acceptance, but I would of course be curious to hear what the other reviewers say.

Below are some more minor/tangential comments/questions:
- The authors frequently make reference to the fact that they are interested in the "pre-asymptotic" regime. To me it would seem intuitive that this would depend quite significantly on the initialization of the approximating distribution, particularly as the divergences and dimensionality are changed. Do the authors have any intuition as to what the best way of initializing the approximate posterior are for these different cases?
- For more complex models, it is often the case that there are symmetries that result in many modes that are identical when it comes to predictions, for instance permuting neurons in the same layer of a neural network would result in such modes. In the case of a unimodal approximate posterior, it would in principle therefore be possible to model one of these modes while missing the others in a way that the predictions are actually reflective of the true posterior. It would seem to me that in such a case the $\hat{k}$ would be quite bad even though the predictions are fine - do the authors have any ideas on how this could be addressed?
- It seems as though the labels are incorrect for Figs. 2 and 3. For instance, in line 138 Fig. 3a is referred to when I believe Fig. 2 is the correct one.
- In Table 1, which approximate posterior is being used?
- l. 259 - "but it less"

**Time Spent Reviewing:**

8

---

> ### Author Response · Authors · 2021-08-10
> **Reply to reviewer**
>
> We thank the reviewer for asking some important questions and their positive review.
>
> Following are the replies to the questions/comments.
> C1: We found that initialization played a more significant role in flow based approximations in contrast to the more restricted variational families, due to the presence of a large number of parameters. We found that initializing the weights such that each flow-transformation resembles the identity function  provided more stable results. However, we made a point of testing the methods in fairly general settings without carefully fine tuning the methods. Initialization is non-trivial for more complex models, and it is difficult to make it general enough to perform well in many scenarios.
> We should also point out that we call this ‘pre-asymptotics’ because the number of samples drawn to estimate the objectives and gradients are not sufficient in high dimensions as reflected by $\hat{k}$, (a safe estimate will need an infeasible amount of samples to be drawn.) This will hold true even after the optimisation has been run for sufficient iterations reducing the effect of initialisation.
>
> C2: We thank the reviewer for raising this point. As we show in our experiments, it is definitely possible to have good performing methods with bad $\hat{k}$ diagnostics. As the reviewer points out, both measures reflect different aspects of the approximating distributions, and it is often a matter of the specific application to trust one or another. We focus on posterior accuracy as it often translates into more general usefulness, and in these cases $\hat{k}$ is a good indicator of how good an approximation is.
>
> C3-C5: We will fix these issues in the camera ready version, we thank the reviewer for pointing these out. Table 1 uses MF Gaussian as the approximating family.

---

> > ### Comment · Reviewer_EXRc · 2021-08-29
> > **Response**
> >
> > Thank you for your response. After reading it along with the other reviews and responses I maintain that this is a good paper worthy of acceptance. However, I note that the authors did not reply to my point about the lack of an in-depth treatment of related work, which was really my main issue with the paper. I highly encourage the authors to consider adding this. Nevertheless, this is a good paper and I look forward to seeing how it turns out.

---

> > > ### Author Response · Authors · 2021-08-31
> > > **related work**
> > >
> > > Apologies for the oversight! We will be sure expand our discussion of prior work. While we borrow some ideas from [14,30], the goal of our paper is quite different: to understand the trade-offs and limitations of different choices for variational objective and approximating family. That is, we aim to provide guidance to end-users *before* they run VI, whereas [14,30] primarily* aim to provide accuracy diagnostics for use *after* running VI. In addition, [14,30] focus on low-dimensional examples while our focus is on approximation high-dimensional posterior distributions.
> > >
> > > *[14] does recommend using heavy-tailed approximations and [30] does discuss simulation-based calibration.

---

### Official Review · Reviewer_8K18 · 2021-07-17

**Rating:** 7
**Confidence:** 4

**Summary:**

The authors propose a conceptual framework and design guidelines for variational inference based on the pre-asymptotic tail behaviour of the unnormalized importance weight. This framework is used to study the behaviour of commonly variational families and f-divergences in mid-to-high dimensions. The authors observe that mass-covering divergences, while often superior theoretically and for low dimensional problems, are hard to optimize in mid-to-high dimensions compared to the mode-seeking exclusive KL-divergence. These observations are consistent with the prediction made using their framework based on estimating the tail-index of the best-fit generalized pareto distribution.

**Limitations And Societal Impact:**

adequate

**Main Review:**

**Originality and Significance:**
As far as I can tell, the proposed framework does not contain novel contributions on a technical level, i.e. the work builds heavily on previous work on pareto smoothed importance sampling (PSIS) [1] and PSIS based diagnostics [2] (as adequately cited by the authors), and the authors use standard estimators for the different f-divergences and their gradients.
That said, I believe there is value, especially for new partitioners, in presenting the above in a coherent and unified framework. The presented framework makes it easy to study the behaviour of commonly used variational families and divergences and helps making educated predictions based on the order of the (estimated) tail index and f-divergence. Hence, I do believe the setup and presentation are original and valuable for the community.

**Clarity:**
The writing of the paper is clear and easy to follow.
One thing that I noticed is that the authors do not explain how to actually use the PSIS correction to improve the estimator. Adding a paragraph explaining this would make the paper more self-contained and enable readers not familiar with [1, 2] to better understand the experiments.

**Quality:**
Overall, the methodology is technically sound and the observations and conclusions drawn by the authors seems correct but sometimes slightly to general.

Specifically, the statements P1, P2 made in section 3.3 are too general to be supported by the actual experiments.
- While *P1* mention 'mode-seeking'-divergences and 'mass-covering'-divergences the experiments only evaluate the exclusive KL-divergence as an example for 'mode-seeking' and the inclusive KL-divergence and $\chi^2$-divergence as examples for 'mass-covering' divergences. The experiments are still insightful, but the claim should be less general to be supported by the actual experiments.
The main text mentions additional results for the other divergences in the appendix, however, the appendix does only include additional results for the evaluation presented in Figure 3c, but not for for the evaluations presented in Figure 3a and 3b.
- The claim made in *P2* would be better supported if the plot in the main text would include the results for the $1/2$-divergence (mentioned on Line 194) and the importance weight $w$ itself. The corresponding plot for Figure 3c can be found in the appendix (Figure B.1.).
   - Is there a reason why these results are not reported?
   - For each of the line plots in Figure 3, what was the variational objective used for training? The figures in the appendix are explicit about what variational objective was used. It would be helpful to mention this in the figure caption in the main text as well.
  - For clarity it would also be helpful to include the degree of the variational objective in the legend similar to the figure in the appendix.

It is also unclear to me if the results are averaged over multiple independently trained models. Given that this paper consideres non-convex stochastic optimization, I consider this necessary to reach any conclusions. Specifically,
- Figure 4, Figure 6 and Table 1 do not report any metric of variability and I can not find any statement in the text mentioning if the presented values are averages over multiple runs.
- How are the confidence intervals computed in Figure 2?
- How is the reported variance computed in Figure 3?

Some clarification would be helpful here.


**Additional Questions and Comments:**
- Regarding the intuition gained by observing the distance from the mode (Figure 1): The overlap between the distribution over distances from the mode is already less for the inclusive KL-divergence in 2D and this difference seems to carries over to the higher dimensional examples.  Even the 2D illustration does not give the impression that the approximation based on the inclusive KL would do better w.r.t. this metric (overlap between distributions over distances). For other metrics, i.e. the variance of the resulting importance weight, I'd agree that the approximation based on the inclusive KL-divergence seems preferable. Overall, I'm not sure how Figure 1 demonstrates that 'the benefits of heavy-tailed approximate families and divergences favoring mass-covering behaviour diminishes as the dimensionality of the target distribution increases'.
Am I missing something here?
- Figure 1; caption: The approximation is red for the exclusive KL and green for the inclusive KL but the caption only mentions that the approximation is shown in red.
- Line 38, 29; some words seem to be missing in that sentence. 'the benefits of heavy-tailed approximate families and divergences favoring mass-covering *behavior* diminishes as *the* dimensionality of the target distribution increases.
- Line 205: I think this should reference Figure 3a instead of Figure 2


*[1] Aki Vehtari, Daniel Simpson, Andrew Gelman, Yao Yuling, and Jonah Gabry. Pareto smoothed importance sampling. arXiv preprint arXiv:1507.02646, 2019.*

*[2] Yuling Yao, Aki Vehtari, Daniel Simpson, and Andrew Gelman. Yes, but did it work?: Evalu- ating variational inference. In Proceedings of the 35th International Conference on Machine Learning, volume 80 of Proceedings of Machine Learning Research, pages 5581–5590. PMLR, 2018.*


**Time Spent Reviewing:**

5

---

> ### Author Response · Authors · 2021-08-10
> **Reply to reviewer**
>
> We thank the reviewer for the thorough review, feedback and useful comments. We will include a section on PSIS. Here, we reply to the comments:
>
> C1: We will include results for ½ divergence to further substantiate our predictions: P1 and P2. That said, however, we find that the growth rate of the f-function is a key defining property of the behavior of f-divergences, and we can generalise the results of our experiments based on this fact. As pointed out in the paper, the exclusive KL divergence depends on the logarithm of the importance weights, whereas mass-covering divergences depend on higher polynomial degrees, which ultimately determine the results.
>
> C2: We thank the reviewer for their suggestion. The main reason to not include 1/2 divergences is clarity and avoiding overloading information in the plots. See reply to C1.
> Each of the lines in figure 3 is the result of fitting the approximation using the variational objective corresponding to the divergence in their label. We will update the text to clarify this point. We will also update the caption to include the degree of the polynomial involved. Some results were left out due to space issues. We will move Fig B1 to the main text.
>
> C3: Results shown in figure 3 and 4 are the average of 50 independent simulations. Regarding Figure 6, we run the experiments with 3 different seeds. We will include standard deviation in table 1. We compute the quantiles in figure 2 as the empirical quantiles in the sample size of size S. In figure 3, the quantiles are empirically computed from 100,000 draws generated from the approximation. We will update the text to include all of this information.
>
> C4: We agree with the reviewer in their assessment. We want to point out that figure 1b is the result of a particular posterior, with fairly low correlation, which explains why exclusive KL behaves that well. In the low-dimensional setting, the inclusive KL does cover the target posterior with a wider and heavier tailed approximation, which is usually a desirable property.
>
> C5-C7:We will update the text and fix these inconsistencies. Thanks for pointing these out.

---

> > ### Comment · Reviewer_8K18 · 2021-08-25
> > **Response**
> >
> > Thank you very much for your detailed answers.
> >
> > After reading the other reviews and responses, it seems to me that the rebuttal addresses most concerns. For my part, I'm happy with the rebuttal and will increase my score, recommending to 'accept' the paper, accordingly.

---

> > > ### Author Response · Authors · 2021-08-30
> > > **response**
> > >
> > > Thank you very much for your reply, comments, updated score and review.

---

### Decision · Program_Chairs · 2021-09-27

**Decision:**

Accept (Poster)

**Comment:**

This paper is concerned with some broad questions in variational inference, namely what variational families should be used (e.g. light-tailed, heavy-tailed, flows) what divergence should be optimized (e.g. inclusive KL, exclusive KL) and how performance could be diagnosed after inference is complete. On a strict technical level, this paper appears to contain few contributions. Nevertheless, reviewers were overall positive about the paper's attempt to establish and support some grand (albeit somewhat informal) themes. These are 1) that mode-seeking divergences are easier to optimize than mode-spanning divergences 2) that this difficulty can be understood by considering the polynomial dependence of the divergence on importance weights and 3) that one can fit a generalized Pareto distribution and use the k statistic to diagnose inference success.

One weakness identified by several reviewers agreed on was an inadequate discussion of prior work. The paper would be stronger with a better review of prior work related to all three of the themes, here, i.e. on the difficulty of optimizing different divergences, on using different variational families for VI and on inference diagnostics. There are some citations now, but with somewhat cursory discussions. In particular, the paper would be stronger if it was self-contained so that someone not familiar with the PSIS framework can follow it. The authors have agreed to expand their discussion of prior work.

Reviewers also had some specific comments about the experimental results (both what was done and the presentation). The authors have also been receptive to this feedback.